# Nanogenerators as a Sustainable Power Source: State of Art, Applications, and Challenges

**DOI:** 10.3390/nano9050773

**Published:** 2019-05-20

**Authors:** Sridhar Sripadmanabhan Indira, Chockalingam Aravind Vaithilingam, Kameswara Satya Prakash Oruganti, Faizal Mohd, Saidur Rahman

**Affiliations:** 1School of Engineering, Faculty of Innovation and Technology, Taylor’s University Lakeside Campus, No. 1, Jalan Taylor’s, 47500 Subang Jaya, Selangor, Malaysia; sridharsripadmanabhannadarindira@sd.taylors.edu.my (S.S.I.); kameswarasatyaprakashoruganti@sd.taylors.edu.my (K.S.P.O.); Mohdfaizal.fauzan@taylors.edu.my (F.M.); 2Research Centre for Nano-Materials and Energy Technology (RCNMET), School of Science and Technology, Sunway University, 47500 Subang Jaya, Malaysia; saidur@sunway.edu.my; 3American University of Ras Al Khaimah, 31291 Ras Al Khaimah, UAE

**Keywords:** piezoelectric nanogenerator (PENG), triboelectric nanogenerator (TENG), pyroelectric nanogenerator (PyENG), self-powered systems, bio-sensors, blue energy

## Abstract

A sustainable power source to meet the needs of energy requirement is very much essential in modern society as the conventional sources are depleting. Bioenergy, hydropower, solar, and wind are some of the well-established renewable energy sources that help to attain the need for energy at mega to gigawatts power scale. Nanogenerators based on nano energy are the growing technology that facilitate self-powered systems, sensors, and flexible and portable electronics in the booming era of IoT (Internet of Things). The nanogenerators can harvest small-scale energy from the ambient nature and surroundings for efficient utilization. The nanogenerators were based on piezo, tribo, and pyroelectric effect, and the first of its kind was developed in the year 2006 by Wang et al. The invention of nanogenerators is a breakthrough in the field of ambient energy-harvesting techniques as they are lightweight, easily fabricated, sustainable, and care-free systems. In this paper, a comprehensive review on fundamentals, performance, recent developments, and application of nanogenerators in self-powered sensors, wind energy harvesting, blue energy harvesting, and its integration with solar photovoltaics are discussed. Finally, the outlook and challenges in the growth of this technology are also outlined.

## 1. Introduction

Energy is an essential requirement for our daily life and sustainable development of our modern society. Energy is required in transportation, communication, aviation, and it also powers our computers, TVs, mobile phones, washing machines, air conditioners, and numerous other electrical and electronic devices. In the past few decades, the rapid industrialization, urbanization, and population growth has increased the demand for energy, leading to the overweening consumption of fossil fuels, which downscales the fossil reserves. The growing global warming, climate change, and diminution of energy reserves led to devoted research towards sustainable and renewable energy sources. As per the 2018 BP (British Petroleum) Energy Outlook, renewable energy is the fastest growing energy source, which accounts for 40% of the increase in energy [1]. Bioenergy, hydropower, solar, and wind are some of the well-established renewable energy sources that help to attain the need for energy at mega to gigawatts power scale. In the past decades, energy-harvesting techniques have drawn the limelight with the expansion of the Internet of Thing (IoT), sensors, wearable, and portable electronics, mobile phones, automatic security systems, defense technologies, etc., because they require power sources that are perpetual and free of maintenance. Generally, such devices rely on batteries for a power source, which has a limited life and environmental hazards. In this regard, harvesting the ambient mechanical energy to power up these devices could be a better choice. Ambient mechanical energy is abundant and the most significant form of renewable energy source among other sources in the environment [2,3,4]. There is a lot of potential in human body motions that can be effectively converted to electricity [5] (Table 1).

This report aims to provide a comprehensive review on piezo-, tribo-, and pyro-based nanogenerators as sustainable energy-harvesting devices including the evolution, development, and applications of the nanogenerators in the past decade. The review is limited to materials used in nanogenerators, the corresponding growth in their electrical output, and their applications in harvesting various ambient energies over the years. The history and general overview of the development of nanogenerators are introduced in Section 2. Then, the working, output power performance, and applications of piezoelectric nanogenerators are summarized in Section 3. 

## 2. History and Development of Nanogenerators

Nanogenerators are generally an energy-harvesting device that generate electricity from waste mechanical energy from the ambient. Earlier, by the end of 17th century, scientists developed practical means to generate electricity from friction. The primitive form of friction machine was developed around 1663 [6]. Later, several researchers worked on this machine to establish the performance. In 1831, the electromagnetic generator was discovered, which is the widely used generator in thermal power plants still today. In 1878, the Wimshurst machine based on static electricity was discovered, which became a popular static electricity generator. In 1929, the famous Van de Graaff generator was invented, which produces very high-voltage direct current. The modern Van de Graaff generators can achieve the very high potential of up to 25 megavolts. Later, technologies were developed to generate electricity based on piezoelectric, triboelectric, and pyroelectric by applying nanotechnology. The high surface area and tunable physical and chemical properties of nanoscale structures promise significantly more efficient technologies that can capture, convert, and store different forms of energy like thermal, radiant, electrical, chemical, and mechanical [7]. The crucial inventions in the history of mechanical energy harvesting are shown in Figure 1a. The energy requirement of various devices at various power scale is represented in Figure 1b. 

So far, there are several ambient energy-harvesting techniques that have been utilized based on the piezoelectric effect, triboelectric effect, pyroelectric effect, and electromagnetic induction, which converts mechanical energy into electricity. Nanogenerators are an evolving energy-harvesting technology that can harvest various classes of mechanical energy such as human motion (i.e., walking, breathing, running, heartbeat), vibration, flowing water, raindrops, and wind; even waste heat can be converted into electricity (pyroelectric effect). The first nanogenerator was developed by Wang et al. in the year 2006 using ZnO nanowires based on the piezoelectric effect, which has a power conversion efficiency of 17–30% [8]. 

There are some available techniques like traditional cantilever-based resonators and transducers to convert mechanical vibrations into electricity but they work effectively for high-frequency vibrations only; on other hand, nanogenerators can be made at nanoscale so that they can effectively harvest the low-frequency mechanical vibrations. In the case of ZnO nanowires, it can be bent more than the bulk ZnO without any damage, making it possible to withstand more strain, and so generates more electricity [8,9]. The different types of nanogenerators and their applications are illustrated in Figure 2. The nanogenerators have shown higher potential and several innovative platforms for mechanical energy harvesting and self-power sensing and monitoring. As technology is growing, there is also a sharp increase in the number of publications in the field of nanogenerators. As per the source of Science Direct database, there are around 469 publications in the year 2018 alone, which is 1.5 times higher when compared to 2017. Figure 3 shows a steady increase in the growth of research in the field of nanogenerators. 

## 3. Nanogenerators Based on Piezoelectric Effect

Piezoelectric nanogenerators (PENG) work on the principle of piezoelectric effect, which means electricity generation when subjected to mechanical stress. In PENG, two electrodes with balanced fermi levels on a piezoelectric material are subjected to an external strain, which creates a piezo potential difference between the internal and external Fermi levels (highest energy state occupied by the electrons) at the contacts [2,8,10,11,12,13,14,15]. To balance this difference in Fermi levels, the charge carriers flow through the external load and a balanced electrostatic level is reached. Alternatively, applying an electric field on a piezoelectric material can cause a mechanical strain. There are two cases of PENG [11,16], one where the individual nanostructure (nanowire/nanorod) [17] is subjected to the strain exerted perpendicular to the growing direction of the nanowire/nanorod, which leads to the generation of the electric field. Figure 4a shows the working of PENG when the force is applied perpendicular to its axis. When a force is applied perpendicular to the direction of the axis of the nanostructure using atomic force microscopy probe, one portion of the nanostructure is stretched (positive strain) while the other undergoes compression (negative strain) [8]. The stretched surface with positive potential was first contacted by the probe, and at this interface, the bias voltage is negative. Thus, a reversed bias Schottky diode is formed with little current. When the probe contacts the compressed side of the nanostructure with negative potential, a biased positive voltage is formed at the interface with sharp peak output current as driven by the potential difference between the two sides. The current flow due to the ohmic contact formed at the bottom of the nanostructure finally re-balances the electric field generated at the tip [8,11]. The conduction is possible only when the top electrode is in contact with the negative potential, whereas no current will be generated if the top electrode is in contact with the positive potential. This is the case for n-type semiconductive nanostructures; in the case of p-type semiconductive nanostructures, it will exhibit the reverse phenomenon since the hole is mobile in this case. The other case is where the external strain is exerted parallel to the growing direction of the nanostructures (Figure 4b). When the force is applied to the tip of the laterally grown nanowire which is stacked between the Schottky contact and ohmic contact, a uniaxial compressive is generated in the nanowire. The tip of the nanowire will have negative potential and increases the Fermi level due to the piezoelectric effect. As the electrons flow from the tip of the nanowire to the bottom through the external circuit, positive potential will be generated at the tip. The Schottky contact blocks the flow of electrons through the nanowires and instead passes the electrons through the external circuit. When the applied force is removed, the piezoelectric effect diminishes immediately and a positive potential at the tip gets neutralized because of the migration of electrons from the bottom electrode to the top, which produces voltage peak in the opposite direction. Due to the in-situ rectifying effect of the Schottky contact, the detected output exhibits direct current characteristics [14]. 

In their work, Zhu et al. replaced this Schottky contact with PMMA layer to create a potential barrier for charge accumulation [18]. In this nanogenerator, when compressive force is applied, a piezopotential field is generated along the nanowires. As a result of electrostatic force, inductive charges accumulate on the top and bottom of the electrodes. This is similar to capacitive configuration in which the strained nanowires can be compared with polarized dipole moments in a plate capacitor filled by a dielectric material. Once the applied stress is released, the piezopotential disappears and the electrons flows back through the external circuit [18,19]. AC output is observed in the cases of capacitive configuration and when the Schottky diode is a series resistance in the piezoelectric nanogenerators. 

To effectively enhance the output power of the PENG, several nanowires are stacked together to effectively synchronize the voltage output of each nanowire. Two effective integrations of nanowires were developed by Wang et al. [11,16]; one is vertical-nanowire-integrated nanogenerator in which the vertically grown nanowires are stacked together (Figure 5a). The working mechanism of vertically integrated PENGs includes lateral bending and vertical compression of nanowires as explained earlier. The other one is the lateral-nanowire-integrated nanogenerator, in which the parallelly grown nanowires are stacked together in the nanogenerator (Figure 5b). In laterally integrated nanogenerators, the deformation of nanowires is always caused by lateral bending either by bending the substrate or by applying pressure on the radial direction of the nanowires [20]. The uniform lateral bending of nanowires can be regarded as the lateral stretching by neglecting the strain distribution in the radial direction due to the ultra-high aspect ratio of the 1D nanostructures. In a study, the energy conversion efficiency of both laterally stretched nanowire and vertically compressed nanowire were compared, and the results showed that the laterally bent nanowire could generate higher voltage than the compressed one. 

### 3.1. Progress and Output Power Optimization in PENGs

The first was developed in the year 2006 based on ZnO nanowires [8,10,11,14]. The aligned nanowires were deflected by a conductive atomic microscope with platinum-coated silicon tip in contact mode. The energy output by one ZnO nanowire (NW) in one discharge event is 0.05 fJ, and the output voltage and power were ~8 mV and ~0.5 pW. For a nanowire density of 20/µm^2^, the output power density is ~10 pW/µm^2^ [8]. The Schottky barrier formed between the microscope metal tip and the nanowires generates power with the power conversion efficiency of 17–30% [8,11,21,22]. Gao and Wang (2007) calculated the piezoelectric potential distribution of a nanowire of 50 nm diameter and 600 nm length as 0.3 V(appx) using perturbation theory [14,15]. The calculation showed that the piezoelectric potential on the surface of the nanowire is directly proportional to the lateral displacement of the nanowire and inversely proportional to the length-to-diameter aspect ratio of the nanowire [15]. 

In 2007, Wang et al. [23] developed a vertically aligned ZnO nanogenerator driven by an ultrasonic wave of frequency 41 kHz, which generated a unidirectional current of 0.15 nA with an open circuit voltage of 0.7 mV and output power volume density of 1–4 W/cm^3^. This voltage is found to be less when compared to the one with an atomic microscope probe as the nanowires are less deflected by the ultrasonic waves. To effectively harvest the mechanical energy, Qin et al. designed a microfiber-based PENG in 2008 using a hydrothermal approach with a ZnO thin film layer as an electrode [24]. This composite structure produces 1–3 mV output voltage and 4 nA current with a power density of 20–80 mW/cm^2^.

Lin et al. (2008) demonstrated the cadmium sulfide (CdS)-based nanogenerator model similar to the ZnO nanowire-based nanogenerator [25]. The nanowire was grown using hydrothermal and physical vapor deposition process. The nanowires produced by physical vapor deposition process seems to produce larger voltage when compared to the nanowires produced by the hydrothermal method. In 2008, Yang et al. fabricated a laterally integrated PENG using flexible substrate without sliding contacts, which are capable of producing alternating current. The fabricated PENG creates an oscillating output voltage (AC) up to ~50 mV when a single nanowire is stretched and released with a strain of 0.05–0.1%. Such type of flexible PENG can be connected in series inside a common substrate to raise the power output [26]. The laterally integrated PENG is effective over vertically integrated PENG [26]. Moreover, the output voltage is 15–100 times higher than direct-current [23] and micro-fiber nanogenerators [24]. Using the same flexible PENG concept later in 2010, Zhu et al. achieved an open circuit voltage of up to 2.03 V, a current of 107 nA, and a power density of ~11 mW/cm^3^. The power generated from this PENG is stored in capacitors and used to light up a commercial light emitting diode (LED). Further, a peak output power density of ~0.44 mW/cm^2^ and volume density of 1.1 W/cm^3^ can be achieved by optimizing the density of the nanowires and by integrating 20 layers of nanowires. [27]. Lin et al. (2008) used light to tune the output performance of the CdS-nanowires-based nanogenerators [28]. The light reduces the height of the Schottky barrier on the nanowires, which gives a positive voltage output.

In 2010, Xu et al. successfully integrated 700 rows of ZnO nanowires to produce a peak voltage of 1.26 v and a maximum current of 28.8 nA at a low strain of 0.19% [29]. Based on theoretical calculation, it is found that within the elastic linear mechanic’s regime, the piezoelectric potential of a nanowire is proportional to the amount of its deformation [15]. So, in a vertical-nanowire-integrated nanogenerator, the nanowires are connected in parallel between two electrodes; as we increase the external strain, their deformation amount increases, growing consequently with the output voltage. The magnitude of the output voltage also depends upon the rate at which the external strain is applied [29]. This high-power output can be used as a power source for neuroprosthetic devices; however, further research is necessary for effective integration. 

Huang et al. (2010) successfully synthesized the first InN (Indium Nitride)-based nanogenerator. The InN nanowire is grown by vapor–liquid–solid (V-L-S) process with the use of an Au nanoparticle as a catalyst [30]. The InN-based nanowire produces both positive and negative piezo-potential, and the maximum reaches up to 1 V, which is highest among all other nanowires. Nanogenerators based on lead zirconate titanate PZT nanofibres were demonstrated by Chen et al. (2010) [31]. The PZT nanofibres were fabricated by the electro-spinning process, and fine platinum wire is used as an electrode, which was assembled on a silicon substrate. The peak voltage and power output were 1.63 V and 0.03 µW; the output voltage depends on the pressure applied to the nanogenerator device. Cha et al. (2011) enhanced the piezoelectric potential by using nanopore arrays of polyvinylidene fluoride (PVDF) and sonic driven [32]; when the input sonic power of 100 dB at 100 Hz, the PENG generates an output of 2.6 V/0.6 µA, which is 5.2 times (piezoelectric potential)/ 6 times (piezoelectric current) higher than the PENG which uses bulk PVDF film, under the same sonic power input.

In piezoelectric materials, due to surface desorption and native defects, free charge carriers are formed [22,33]. Lu et al. (2012) found that these free charge carriers affect the piezoelectric potential known as the screening effect [34]. ZnO nanorod was used for the study and they were illuminated using UV light; the carrier concentration increases up to 5.6 × 10^18^ cm^−3^ under 1.2 mW/cm^2^ illumination. As the UV light intensity increases, carrier concentration also increases, which makes the current-voltage characteristics insensitive. The carrier concentration can be reduced by improving the intrinsic properties using surface passivation, thermal annealing, and oxygen plasma [35,36,37,38]. Pham et al. (2012) applied a simple thermal annealing treatment to the pristine ZnO nanorods in the presence of UV light and found the output piezoelectric potential was 25 times higher [39].

Later in 2012, Zhu et al. demonstrated vertically integrated position-controlled piezoelectric ZnO nanowires that convert biomechanical energy into electrical energy with a high-level open-circuit voltage of 58 V, short circuit current of 134 µA, and a maximum power density of 0.78 W/cm^3^ [18].

Hu et al. (2012) improved the performance of the nanogenerators by using pretreatment methods like oxygen plasma, annealing air, and surface passivation with specific polymers on the grown ZnO nanowire films. The nanogenerator’s output voltage reached 20 V and the output current exceeded 6 µA [35]. This nanogenerator successfully powered an automatic watch for more than 1 min (for 1000 cycles of deformation of nanogenerator). 

Zhou et al. (2012), for the first time, demonstrated the energy-harvesting potential and piezotronic effect in vertically aligned CdSe nanowire arrays [40]. Platinum is used as an electrode when a reasonable force or stress is applied on the nanowire with a maximum output voltage of 137 mV. The Schottky barrier between the platinum and the CdSe reduces the current. 

Han et al. (2013) presented an innovative three-dimensional r-shaped hybrid NG design based on piezoelectric and triboelectric energy harvesting [41]. The output performance of the device was enhanced by fabricating micro- or nanoscale devices on a polydimethylsiloxane (PDMS) surface, which was placed under an aluminum electrode on PVDF. The Al electrode was shared in common by both the piezoelectric and the triboelectric component. The piezoelectric and triboelectric generators exhibited an increased power density of 10.95 mW/cm^3^ and 2.04 mW/cm^3^, respectively. The hybrid r-shaped design showed relatively high reliability, as its performance was not degraded over 6000 continuous cycles under an external force with a frequency of 10 Hz. 

Lithium-doped ZnO nanowires are used in large-scale nanogenerators for high performance [38,42]. Lu et al. (2015) [43] utilized Au particles on the surface of the ZnO to reach an output voltage of 2 V and a current density of 1 µA/cm^2^.

Ghosh and Mandal (2016) highlighted the intrinsic piezoelectric property in transparent fish scale, which is composed of self-assembled and ordered collagen nano-fibrils, and serves as a self-poled piezoelectric active component with a piezoelectric strength of −5.0 pC/N [44]. A robust nanogenerator is fabricated by using gold electrodes of 90 nm thickness on both sides of the fish scale by sputtering followed by lamination with polypropylene film. This type of bio-piezoelectric nanogenerator under the repeated compressive stress of 0.17 MPa generates an output voltage of 4 V, short circuit current of 1.5 µA, and maximum output power density of 1.14 µW/cm^2^. An enhanced output voltage of 14 V was obtained by serially integrating four of these bio-piezoelectric nanogenerators. 

HaiBo et al. (2017) found that polymorphic phase sodium-potassium niobate (NKN) nanorods have the most significant piezoelectric strain constant (175 pm/V) as they have more directions for dipole rotation than the nanogenerators with rhombohedral or orthorhombic nanorods [45]. Their experiment, using 0.7 g of PP (polymorphic phase) NKN nanorods, showed a maximum open-circuit voltage of 35 V and a short circuit current of 5.0 at a strain of 2.13% and an average strain rate of 3.7% s^−1^. This nanogenerator generates a maximum power output of 16.5 µW for a load of 10 MΩ. 

Chen et al. (2017) proposed a flexible piezoelectric nanogenerator based on a vertically aligned nanocomposite micropillar array of polyvinylidene fluoride-trifluoroethylene (P(VDF-TrFE))/barium titanate (BaTiO_3_), which exhibits an enhanced voltage of 13.2 V and a current density of 0.33 µA/cm^2^ [46]. Upon the application of a force of 3 N at a frequency of 5 Hz, a flexible PENG based on barium titanate embedded polyvinylidene difluoride (i.e., BaTiO_3_/PVDF) composite film exhibited a high output voltage of 14 V and short circuit current of 0.96 µA [47]. 

Shi et al. (2018) fabricated a PENG by using electrospun nanocomposite fibre mats composed of 0.15 wt% graphene nanosheets and 15 wt% barium titanate nanoparticles, which generates a steady electric power of 11 V and 4.1 µW at a load frequency of 2 Hz and a strain of 4 mm even after 1800 cycles [48]. The PENG also generates a peak voltage of 112 V during a finger pressing–releasing process, which is capable of powering 15 LEDs and a watch. 

Jenkins et al. (2018) explored the behavior of diphenylalanine peptide using finite element analysis and found that this peptide nanowires can generate significantly higher power output than the nanowires made up of ZnO, lead zirconate titanate, and barium titanate [49]. The nanogenerator made out of this nanowire achieved an open-circuit voltage up to −0.6 V and short circuit current up to 7 nA. The output voltage remains stable for more than 1000 cycles and the maximum power generated was 0.1 nW for a load resistance of 100 MΩ. Recently, in PVDF-based piezoelectric nanogenerators, the performance is increased by fabrication techniques, piezoelectric materials, conductive, and non-conductive fillers, which increases the piezoelectric crystal structures, alignment of dipoles, and charge transfer [50]. The innovative 3D core multishell PENG showed improved performance [51]. 

Kang et al. used GaN nanoporous layers instead of nanowires, using electrochemical etching process in their PENG [52]. This suppresses the carrier screening effect and enhances the output voltage. Kang et al. (2017) demonstrated the transfer of a large area GaN membrane onto a flexible PET substrate in a transparent flexible piezoelectric nanogenerator using electrochemical lift-off process resulting in an output voltage and current of 4.2 v and 150 nA [53]. The same electrochemical lift-off process was used by Johar et al. to fabricate a flexible PENG by forming a p-n NiO/GaN heterojunction [54]. The lift-off process removes the residual stress in the GaN layer, which suppresses the free carrier screening. The developed PENG is capable of harnessing energy from the airflow, finger forces, and vibrations at a frequency of 20 Hz. Johar et al. (2018) fabricated the GaN (Gallium Nitride)-based piezoelectric nanogenerator using Ni as contact metal [55]. The output performance was enhanced by using polydimethylsiloxane (PDMS) as a dielectric medium between GaN nanowire and Ni electrode. A maximum output voltage and current of 15 V and 85 nA were generated. The significant improvements in the development of PENG over the years are illustrated in Table 2.

It is evident from Table 2 that the condition of deformity, output power, and output power density was not always mentioned, which is essential to compare the performance among the PENGs. In all the studies, the authors have highlighted the output voltage obtained for the large surface, which is quite impressive to present the output results.

### 3.2. Applications of PENGs

PENGs acts as a sustainable power supply for various smart applications like self-powered nano/microsensors, self-powered electronics, wearable/flexible electronics, and biomedical applications [2,13,14,20,22,59,60]. 

Hu et al. (2011) [61] integrated the ZnO nanowire-based PENG onto a tire’s inner surface; the deformation of the tire during rotation gives a power output of 1.5 V and 25 nA with a maximum power density of 70 µW/cm^3^. The PZT nanofibres have higher piezoelectric voltage constant and dielectric constant, making it ideal for nanogenerator and nanobattery applications [31]. ZnO nanowire-based biosensors were developed in 2014, which paved the path for the future development of biosensing devices [62]. PENGs effectively harvest energy from the motions of internal body parts and powers health monitoring devices and implantable devices like pacemakers, cardioverter-defibrillators, and neural stimulators [63,64,65]. ZnO nanowires, indium tin oxide (ITO), and PZT film-based flexible and transparent nanogenerators were developed to harness power from finger typing [66,67,68]. Piezoelectric nanogenerators that have excellent flexibility and high output voltage have promising applications in power electronics.

Various hybrid energy harvesters that integrate PENG with other types of energy harvesters like triboelectric nanogenerator (TENG) and pyroelectric nanogenerator (PyENG) were developed for ubiquitous power generation and improved power conversion efficiency [69,70,71,72,73,74]. PENGs were also used in solar PV cells for improved power conversion efficiency. A tandem nanogenerator was developed by integrating silicon nanopillar solar cell with PVDF nanogenerator; this device was capable of harvesting energy from both sound waves and solar energy [75]. Zhu et al. (2017) fabricated a silicon-based nanoheterostructure photovoltaic device, which is based on the piezo-phototronic effect. The efficiency of the solar cell was improved from 8.97% to 9.51% [76].

Chemically reinforced composite-based PENG produces a maximum AC voltage of 65 V, which is converted into DC output using a filling wave bridge rectifier to charge capacitors and power LEDs [57]. Maity and Mandal (2018) designed an organic piezoelectric nanogenerator based on multilayer structure of PVDF NFL mats, followed by PEDOT coating, which exhibits an open circuit voltage of 48 V under the stress of 8.3 kPa [77]. This power output suggests its application in the field of self-powered wearable and portable electronics. An inorganic–organic hybrid piezoelectric nanogenerator based on zinc sulphide nanorods and electrospun PVDF (polyvinylidene fluoride) possesses a resonance frequency of 86 ± 3 Hz, an acoustic sensitivity of ~3 V/Pa, and very high wind energy conversion efficiency of ~58% [78]. This makes it capable of noise detection, wind energy harvesting, security monitoring, and also useful in self-powered sensors. There are several security systems sensors like transport monitoring [79], wireless sensors [80], and biomedical sensors [79,81] based on PENGs, which holds practical importance.

### 3.3. Outlook on PENGs

The PENGs have high output performance when compared to the other piezoelectric energy-harvesting techniques. The ZnO-based flexible nanogenerators were able to produce power output 11–22 times higher than the PZT-based bulk cantilever energy harvester [27,29]. The multifunctional piezoelectric nanogenerators are the exact source of power for wearable and implantable devices. As these nanogenerators are integrated with electronics, dresses, and human bodies, future development should be focused on flexibility, durability, and stability. Organic polymers with high flexibility have to be identified to replace the existing organic polymers. The research on semiconducting nanowires is crucial to further improve the performance and the applications of PENGs. The first PENG was based on ZnO nanowires, and there were several improved models with ZnO nanowires. Apart from these, various other 1D nanomaterials like CdS, GaN, ZnS, InN, CdSe, InAs and 2D MoS_2_ that have good piezoelectric potential were also studied [25,30,31,53,55,82,83]. Optimization of structural design, integration, and packing of nanogenerators for self-powered sensors are some of the future requirements that have to be addressed for efficient electromechanical energy conversion. 

## 4. Nanogenerators Based on the Triboelectric Effect

Triboelectric nanogenerators (TENGs) work on the combined mechanism of triboelectrification and electrostatic induction. This device was invented accidentally when Zhong and his team fabricated a piezoelectric nanogenerator with a little gap that showed a high output voltage; later, it was found that the high output was due the triboelectric effect [84]. Triboelectrification refers to charge generation on the surface of two dissimilar materials when they are brought into contact. Electrostatic induction is an electricity-generating phenomenon in which electrons flow from one electrode to another electrode through an external load to equalize their difference in potential [85,86,87,88,89,90]. In TENGs, as the two dissimilar dielectric materials have a different surface, electron affinities are brought into contact and the time-varying triboelectric charges are formed on their surfaces, and when they are separated, an electric potential is formed between them. The mechanism behind the formation of charges on the surface of the triboelectric material was not defined clearly and remain highly debated [89,91]. A fundamental physical model was formed only in the year 2017, in which the triboelectrification mechanism that has been known for thousands of years was traced back to Maxwell’s Displacement current. The materials that show the triboelectrification phenomenon are usually insulators and have the ability to retain the transferred charges for a long period of time, and because of this nature, there are many negative effects caused in industrial manufacturing, transportation, aviation, etc. Back in the late 18th century, the first triboelectric-based generator named the Wimshurst machine (~1880) was built. Later in 1929, the Van de Graff Generator was invented, which is used for generating high voltage [89]. Later, the triboelectric-based nanogenerator was first invented in the year 2012, which harvests ambient mechanical/vibration energy [89,92,93,94]. There are four unique elemental modes of operation of TENGs so far, including (i) vertical-contact separation mode, (ii) in-plane sliding mode, (iii) single electrode mode, and iv) freestanding triboelectric layer. The comparison of the working principle of all the modes of operation is compared in Table 3. The triboelectric materials, based on their ability to gain electrons and lose electrons, were arranged in a series called triboelectric series, published by John Carl Wilcke in 1757 [95] (see Table 4). Traditionally, materials like silk, wool, nylon, and polymers like polytetrafluoroethylene (PTFE), PVDF, and PDMS were used as a triboelectric friction layer, but the choice of materials for the electrode is unlimited [95,96,97].

### 4.1. Progress and Output Power Optimization in TENGs

The first flexible triboelectric nanogenerator was invented in the year 2012 by Wang’s group by sandwiching polyester (PET) and Kapton thin films, which generated an open-circuit voltage of 3.3 V and current of 0.6 µA at a power density of ~10.4 mW/cm^3^ [88]. Later in the same year, Zhu et al. gave a proper explanation for the TENG-based power conversion utilizing polymethylmethacrylate (PMMA) and Kapton as triboelectric materials [98]. These new types of robust nanogenerators produce a maximum open circuit voltage of 110 V, and the instantaneous power density reached up to 31.2 mW/cm^3^. Wang et al. (2012) designed an arc-shaped triboelectric nanogenerator using polymer thin film and a thin metal film and studied the working using finite element analysis. The designed arc-shaped generator [98] reached a power output of 230 V, 15.5 µA/cm^2^, and 128 mW/cm^3^. 

Polydimethylsiloxane is considered one of the suitable materials for TENG applications because of its distinct properties like flexibility, transparency, high negative polarity, and easy fabrication. Zhu et al. (2013), based on contact separation mode, developed a TENG using nanoparticle-enhanced polydimethylsiloxane (PDMS) and gold (Au) thin film, which reached a record high voltage of ~1200 V, instantaneous power output of 1.2 W, power density of 313 W/m^2^, with an average output of 132.1 mW [99]. Yun et al., in their work, exposed the PDMS to ultraviolet-ozone and then sprinkled with NaOH solution to get a triboelectric voltage of 49.3 V and 1.16 µA, respectively [100]. This output is found to be 15 times larger than the TENG using fresh PDMS. Hu et al. (2013) designed a vertical contact TENG integrated with a 3D spiral structure that gave a maximum power output of 2.76 W/m^2^ on a load of 6 MΩ at a resonant frequency of 30 Hz [101].

A harmonic-resonator-based TENG has been designed as an active vibration sensor, which was coupled with nanomaterials modifications. When a vibration with frequency from 2 to 200 Hz with the considerable working frequency of 13.4 Hz is applied, it produces a uniform quasi-sinusoidal output of 284.7 V, a current of 76.8 µA, with a peak power density of 726.1 mW/m^2^ [102].

Usually, a nanogenerator generates high output voltage and low output power, so in order to improve the instantaneous power output of the TENG, Cheng et al. (2013) devised a nanogenerator based on an off–on–off contact switching during the mechanical triggering that largely reduces the duration of charging/discharging process, so the pulse of the instantaneous power output improves without compromising the output voltage. The output current and voltage reaches as high as 0.53 A and 142 W at a load of 500 Ω [103]. The power current and power density reach as high as 1325 A/m^2^ and 3.6 × 10^5^ W/m^2^, respectively. 

As a solution for lower power output in TENGs, 3D integrated multilayered TENGs were designed later in 2014 by Yang et al. In this type of TENG, the output of individual TENGs are synchronized to achieve a maximized instantaneous power output [104]. This 3D TENG has a multilayered structure with acrylic supporting substrates. PTFE nanowires are used as triboelectric materials, and aluminum and copper are used as electrodes and contact surfaces. The synchronized 3D TENG produces a short circuit current of 1.14 mA and an open-circuit voltage of 303 V with a power density of 104.6 Wm^−2^, which is capable of lighting 20 spotlights (0.6 W each) and a white G 16 globe light. 

Lin et al. [105] studied the relationship between the motion of a single water drop on to the water TENG and proposed a sequential contact-electrification and electrostatic-induction process to understand the working mechanism of it. The TENG is made out of nanostructured PTFE thin film and PMMA as substrate layer coated with Cu electrode. The TENG is based on a single electrode mode. When a 30 µL water drop hits the TENG, it can achieve a peak voltage of 9.3 V and a peak current of 17 µA. A maximum power output of 145 µW is obtained when the generator is connected to a resistor of resistance 5 MΩ. The study also proved that the superhydrophobic nature of the PTFE film is responsible for the high output.

Yong et al. [106] demonstrated a wind-rolling triboelectric nanogenerator that generates electricity from wind as a lightweight triboelectric sphere rotates along a vortex whistle substrate. A single unit of this TENG generates an open-circuit voltage of 11.2 V and closed-circuit current of 1.86 µA. The output power is also enhanced through multiple electrode patterns as well as by increasing the number triboelectric spheres inside the nanogenerator.

In 2017, Chen et al. came up with an ultra-flexible 3D TENG manufactured by unique hybrid UV 3D printing [107]. The TENG makes use of printed composite resin parts and ionic hydrogel as an electrification layer and electrode. A decent output of 10.98 W/m^3^ and 0.65 mC/m^3^ is obtained at a low frequency of ~1.3 Hz.

Mallineni et al. (2018) successfully harvested mechanical energy via a 3D printed wireless triboelectric nanogenerator using graphene polylactic acid nanocomposite and Teflon as triboelectric surfaces [97]. When actuated by simple hand motions, this TENG generated a record high output voltage of 2 kV with instantaneous peak power up to 70 mW. This steady voltage output enables the transmission of the electric field over a distance of 3 m wirelessly to charge a capacitor.

The improvement of TENGs output performance has always been a challenging issue in the progress of TENG. Theoretically, the energy conversion of TENG is 85%, but so far in practice, it has not yet been achieved [96,108]. Designing a new working mode and enlarging the frictional triboelectric surface area and the frequency can significantly influence the output performance of the TENGs [109]. Based on this, Feng et al. (2018) came up with an innovative TENG based on biodegradable leaf and leaf powder, which gives 15 µA current and 430 V under 5 Hz contact mode [96]. In the same device, poly-L-lysine (PLL) is applied to modify the leaf powder, which enhances the output performance as high as 60 µA and 1000 V. A conductive double-sided carbon tape composed of carbon powder was proposed as electrodes for TENGs by Shi et al. [110] in 2018 for higher output power density. PDMS and polyamide 6 were used as a tribo-frictional layer, owing to a strong interaction between the carbon electrode, and the tribo-layer demonstrates a peak output voltage of ~1760 V, short circuit current of ~240 mA/m^2^, and power density of ~120 W/m^2^. The output performance is found to be much higher than TENGs based on aluminum electrodes. It was also found that when the polyvinyl chloride was treated with CF_4_, the performance of TENG was increased [111]. A newly designed TENG with integrated rhombic gridding improves the power output performance, which is based on the hybridization of both contact separation and sliding electrification mode [112].

Piezo-hybrid-enhanced triboelectric nanogenerators fabricated by electrospinning silk fibroin and PVDF nanofibres showed outstanding electric output performance with a power density of 3.1 W/m^2^ [113]. Inspired by the principle of lightning rods, utilizing electrostatic discharge to improve the TENGs performance was first proposed by Zhai et al. [114]. The different needle structure is used to produce the electrostatic discharge and transform the enormous potential into free charges. To maintain the stability and reliability of the electrostatic discharge process, a set of microdisplacement platform and argon encapsulation technology is used during the assembly.

A TENG has been developed by mimicking the characteristics of kelp to harness wave energy [115]. They are made of flexible materials and consist of vertically free-standing polymer strips. Every single strip could sway independently to cause a contact–separation with the neighboring strips when the TENG vibrates in waves. An output current of 10 µA and voltage of 260 V can be given by the single unit with a power density of 25 µW/cm^2^. Control of dielectric constant fluorinated polymers [116], chemical modification of polymer surfaces (tribo-layer) [117], multilayered fiber-based TENG [112,118], and grating structures [108,119] were used in the TENG to enhance the output performance.

In 2018, researchers demonstrated flexible triboelectric nanogenerators using a new family of two-dimensional layered transition metal carbides and nitrides called metallic MXene [120,121,122,123]. The most widely used material in manufacturing TENG is PTFE (polytetrafluoroethylene) but PTFE-based TENGs are often restricted to single electrode mode-based operation, and also its property of high thermal and chemical stability and lower surface energy hinders the easy fabrication of metallic coating over it. The MXene materials have the ability to inherent the conductivity as well as the other limitations of PTFE in fabricating TENGs. Dong et al. [120] demonstrated a flexible MXene TENG capable of harvesting electricity from the motion of human muscle movement. MXene was used as the triboelectric material, and the PET-ITO layer was used as the electrode in the TENG. The study also proves that the MXene-based flexible TENG produces high open circuit voltage ranging between 500–600 V with an instantaneous peak power output up to 0.5–0.65 mW. Jiang et al. [121] developed a flexible single electrode mode TENG with integrated MXene-based micro-supercapacitors. The supercapacitors are used to store the energy in standby mode and supply power when active. The MXene-based micro-supercapacitors supplies a capacitance of 23 mF/cm^2^ with 95% capacitance retention after 10,000 charge–discharge cycles. This helps the TENG to achieve a maximum output power of 7.8 µW/cm^2^. These types of TENGs opened new possibilities in wearable and implantable sensors networks.

A direct current TENG was designed without using a rectifier by Luo et al. in 2019 [124]. This TENG functions by using air break down, which was once a negative effect in TENGs [125,126,127]. This type of DC-TENG is used in making flexible nanogenerators because of its simple structure. Kang et al. [128] in 2019 systematically studied the surface charge potential of metal nanowire-polymer matrix hybrid layer and its effects for the performance enhancement of TENG. Through experimentation, it has been found that increasing the density of the metal nanowire embedded in the polymer matrix resulted in the transfer of more charges from metal nanowires the polymer film.

The efficiency of TENG drops as the temperature increases [129] and this challenge of operating TENG at high temperatures was addressed by Xu et al. [130]. The temperature of the TENG was raised to 673 K, and the dominant deterring factor thermionic emission is prevented by using preannealing. This study improves the application of TENGs in high-temperature environments like outer space, jet engine nozzles, and geothermal energy.

### 4.2. Review of Applications of TENG

#### 4.2.1. Self-Powered Sensors

There are wireless TENGs with very high output performance that have applications in actuating smart home application like lights, temperature sensors, burglar alarms, smart windows, garage doors, etc., [97]. The development of TENGs contributes largely to the current expansion of the Internet of Things (IoT), artificial intelligence (AI), autonomous robotics, virtual reality, and human–machine interfacing ecosystem [131]. The TENG-based intelligent keyboard [131,132], touch screen [133], E-skin [134,135], acoustic sensor [136], and artificial muscle [137] can be a strong encomium and a paragon shift in the existing human interface ecosystem.

TENG made out of polyamide 6,6 (PA) film or polytetrafluoroethylene (PTFE) film, is applied to an active sensor for detecting water or ethanol in gas or liquid phase [138]. The authors [139] designed a flexible cylindrical spiral type TENG (S-TENG) and a self-powered measuring tapeline (sensor) based on the TENG that was capable of generating open circuit voltage up to ~250 V. The S-TENG with such an enhanced power output can power display devices and sensors, thus eliminating the need of batteries. The 3D stacked TENG is successfully integrated inside a ball (basketball, football, baseball) of diameter 3 inches to harvest wasted kinetic energy when people play ball sports [104]. The experimental study showed a constant voltage of 5 V even after 3000 working cycles; hence it can be applied in self-powered sensors. The authors [140] proposed an on-vehicle magnetic triboelectric nanogenerator mounted on the wheel hub of a tire for self-powered tire pressure monitoring sensors, barometric pressure sensor, and wireless sensor nodes. This type of TENG can deliver an open circuit voltage of 316 V at the rotation speed of 100 rpm. TENGs integrated with vehicle tires convert frictional energy into electricity and also harness the static electricity, which hinders the wide-scale application of silica-filled green tires [141]. The newly fabricated DC-TENG can find its application in flexible electronics and self-charging power systems [124].

TENGs were successfully integrated with the shoes and backpacks to harvest the human walking energy [107,142]. Liu et al. developed a hybridized nanogenerator by combining a TENG and an electromagnetic nanogenerator, which can be embedded into wearable shoes of soldiers as an energy cell [111]. On the other hand, TENG was successfully integrated into a backpack, which can harvest vibration energy from human walking [112]. This device can act as a mobile power source for engineers and explorers. Zhong et al. (2012) demonstrated an arc-shaped flexible nanogenerator that can generate power from finger typing. The output power of the TENG device reached as high as ~4.125 mV, which is sufficient to light 50 commercial LEDs connected in series. This proves its application in the self-powered system and sensors [143]. Wearable and stretchable TENGs based on Kinesio tape exhibits a linear relationship with the stretched displacements and bending angles, enabling it to harvest energy from human motions like knee joint bending and human gestures [144]. Apart from these, there are TENGs that harvest electrical energy from eyeball motion [145], eye blinks [146], human body gestures, and posture change [147].

#### 4.2.2. Harvesting Blue Energy

Water wave energy is one of the widely distributed and promising source of ambient energy [148,149], which is conventionally harvested using electromagnetic generators (EMG). The newly developed TENGs were capable of harvesting this large-scale blue energy. The main advantage of TENG over EMG is that they can harvest energy even in the irregular environment and at a low frequency of less than 5 Hz [148].

TENGs that harvest liquid wave energy, in which water itself acts as one of the triboelectric materials, have been developed [150,151]. The recurrent contact between water and the dielectric material (PDMS, FEP) layer causes electron flow across the external circuits [151]. Generally, water causes charge buildup in hydrophilic particles [152] and metals [153]. It also can hold charges for long periods [154]. Water is located on the top of the triboelectric series and it is always a positive charge, but it acquires a negative charge when falling through the air [155].

By integrating a 3D spiral structure, a TENG inside a buoy ball energy from wave can be harvested to give a power output of 110 V and 15 µA current [101]. This type of TENG has its application in marine sciences and environmental science for powering the sensors. A spherical TENG of 6 cm diameter actuated by water waves can generate a peak current of 1 µA and instantaneous power of 10 mW [156]. Kapton and Nylon 6/6 were used as the triboelectric surfaces based on freestanding triboelectric layer mode. These rolling TENGs [156,157,158] have a simple structure, are lightweight, and capable of rocking on or in water to harvest large-scale wave energy in lakes and oceans. Based on the output of a single TENG, a network of TENGs that can naturally float on the water surface and convert the low-frequency wave energy into electricity was developed [159]. This type of network of TENGs [160] is expected to produce an average power output of 1.15 MW from 1 square kilometer surface area. Zhong Lin Wang proposes a blue energy dream, in which thousands of TENGs are linked into a network using cables that could power a town or purify saline water [160]. This network can also be hybridized along with solar panels and wind turbines that could power an electricity grid. The network of TENGs, when integrated with the power management module, could produce a constant DC voltage on the external load and can be stored in capacitors. Liang et al. integrated a network of TENGs with a power management module, which improves the stored energy by 96 times in charging a capacitor [149].

Concerning low toxicity, excellent biocompatibility, and biodegradability, Pang et al. fabricated a TENG using alginate as a triboelectric layer [161]. A butterfly-inspired B-TENG was proposed by researchers to harvest multidirectional ocean wave energy [162]. The spring-assisted four-bar linkage with four double-faced copper-clad plastic sheets in the B-TENG help to move the TENG in many directions to efficiently harvest energy. A recently developed TENG (2019) efficiently collects water wave energy and powers a self-activated anti-biofouling system in marine systems [163]. The researchers came up with optimized wavy structured [164,165,166,167] and nanowire [168] based TENGs that showed performance improvement in blue energy harvesting. In addition to these, there are other structures like duck-shaped [169], air-driven [169], membrane and rolling rod [170,171,172,173,174], and box type-based [175] TENG for efficient blue energy harvesting.

#### 4.2.3. Harvesting Wind Energy

Wind energy is one of the promising sources of renewable energy source that requires alternative harvesting methods. Researchers developed a wind-rolling triboelectric nanogenerator that generates electricity from wind as a lightweight triboelectric sphere rotates along a vortex whistle substrate [106]. An angle-shaped TENG overcame the problem of insufficient contact of the triboelectric layers with enhanced power output [176]. A farm structure TENG harvests low wind energy into electrical energy [177]. Other than these, the leaf-based [96] and free-standing flag type [178] TENGs can be used to harvest wind energy in smart cities [179].

#### 4.2.4. Solar Photovoltaic Technology

A highly transparent triboelectric nanogenerator has been developed by Liang et al. to harvest electrostatic energy from flowing water. The TENG is made up of a fluorine-doped tin oxide (FTO) electrode and a PTFE film. For high transparency, the PTFE film was prepared at a thickness of less than 1 µm. The antireflection coating is also used in the TENG to increase the transmittance so that they can be integrated into solar cells, building glass, and vehicle glass. Upon the impact of water, this transparent TENG produces a peak output voltage of 10 V and a current density of 2 µA/cm^2^. The output power density reached 11.56 mW/m^2^ when connected to 0.5 MΩ resistors.

Later in 2014, Zheng et al. [180] successfully integrated the transparent TENG in solar cells by replacing the existing protection layer without affecting the performance of the first solar cell. The TENG is made out of a specially processed superhydrophobic PTFE thin film, an indium tin oxide (ITO), and a polyethylene terephthalate (PET) layer. The superhydrophobicity of PTFE film enhances the automatic cleaning of the solar cell surface. The experiment showed that when the dripping water rate is adjusted at 0.116 mL/s due to the sequential process of contact electrification [181] and electrostatic induction, the TENG produces an open circuit voltage of 30 V and a short-circuit current density of 4.2 mA/m^2^. The experiment also showed that as the rate of water dripping increases, the magnitude of the current peaks decreases while the density of the current peaks increases. A water drop TENG with a surface area of 614 m^2^ can light a lamp of 10 W power.

Liu et al. integrated a silicon solar cell with triboelectric nanogenerator using a mutual electrode for harvesting energy from both sunlight and raindrops [182]. A heterojunction silicon solar cell is used, and polydimethylsiloxane (PDMS) is used as a triboelectric layer combined with PEDOT: PSS as an electrode. The imprinting pattern of the DVD was transferred onto the PDMS film and PEDOT: PSS layer which helps in achieving a low reflection ratio that enhances light harvesting without affecting the output voltage and power conversion efficiency. This integrated system converts both solar energy and raindrops into electrical power with higher outputs of ~33 nA and 2.14 V with a maximum average power density of 1.74 mW/m^2^.

For the first time, Zheng et al. in 2015, developed a hybrid solar panel that simultaneously generates power from sunlight, raindrops, and wind using TENG [183]. A waterproof fabric-based multifunctional hybrid triboelectric nanogenerator was developed recently to harvest energy from raindrops, wind, and human motions [184]. This newly developed TENG is as flexible as clothes, making it suitable for wearable technologies. These types of TENGs changes the traditional way of solar energy harvesting and provides a new vision for all weather solar photovoltaic technologies. High humidity and contamination-resistant triboelectric nanogenerators with superhydrophobic interface [185] can be sandwiched with solar PV cells so that the PV cell can be dust resistant. Significant improvements in the TENGs are illustrated in Table 5. In Table 5, the outputs achieved in TENGs over the years were compared without considering the size of the nanomaterial or the quantity of force applied, which also determines the output power.

### 4.3. Outlook on TENGs

The comparative table shows the triboelectric nanogenerators with various materials and their respective power output. The maximum voltage output obtained through research over the years in the field of triboelectric nanogenerators is shown in Figure 6. The trendline shows a linear improvement in the output voltage of TENGs over the years. Apart from the choice of materials from the triboelectric series, by increasing the frictional force between the triboelectric surfaces, the output performance of the TENG can be enhanced. The output performance of the TENGs was also largely dependent upon the contact materials used. The contact materials can be made out of composites by embedding nanoparticles in a polymer matrix; by doing this, the electrostatic induction can be enhanced by altering the surface electrification and permittivity. Techniques like forming pyramid-, square-, and hemisphere-based nanopatterns over the triboelectric surfaces improves the contact area, thus improving the triboelectrification. This gives a more significant opportunity for the researchers to improve the efficiency of TENGs through extensive research at the material level. The DC TENGs opens a new opportunity for research development in DC energy harvesting and flexible electronics. Moreover, when compared with PENGs, the output voltage of TENGs are very high and hence has a wide range of applications. This endorses a more significant number of research developments in the field of TENG, unlike PENG.

## 5. Nanogenerators Based on Thermoelectric Effect (Seebeck Effect)

Thermoelectric generators (TEG) function like heat engines, which convert the temperature difference into electric voltage. The process of direct conversion of the temperature difference into electric voltage is called a Seebeck effect [200]. The same process can be reversed, that when an electric current is passed across a junction between two different conductors, heat is either absorbed or is produced on the junction. In TEG, the generated voltage is directly proportional to the temperature gradient [201,202] (i.e., temperature changing over distance).

The thermoelectric generators are most commonly used as a micro energy harvester to harvest waste heat energy in power plants, the engine of moving automobiles, solar panels, CPU of computers, the human body, etc. [200,202,203]. To efficiently harvest the waste heat energy using the Seebeck effect, the temperature gradient should be very high [204] also the thermoelectric materials are costly. The first thermoelectric nanogenerator was developed in 2012 by Wang’s group based on a single Sb-doped ZnO nanobelt [203]. The single Sb-doped ZnO micro belt shows a Seebeck coefficient of about −350 µV/K and a high-power factor of about 3.2 × 10^−4^ W/m K^2^. A single Sb-doped ZnO micro belt under a temperature difference of 30 K produced an output voltage of 10 mV and an output current of 194 nA. This type of nanogenerators can be used in self-powered temperature sensors. In the year 2016, Rojas et al. [205] developed a paper-based and flexible thermoelectric nanogenerator based on paper substrates using simple micromachining and microfabrication techniques. A higher power of ~80 nW was achieved when a smooth polyester paper is used at a temperature difference of 75 K. Such a type of nanogenerator are promising power sources for flexible and wearable electronics.

## 6. Nanogenerators Based on Pyroelectric Effect

Pyroelectric nanogenerators are marvelous energy-harvesting devices of the future that have the enormous capability of converting thermal energy to electric energy by utilizing nano-sized pyroelectric materials. The main difference between the Seebeck effect-based thermal energy harvesters and pyroelectric generators is the time-dependent temperature. In the Seebeck effect, we cannot generate power from time-dependent temperature fluctuation because spontaneous polarization is not possible in the Seebeck effect. The first pyroelectric nanogenerator was developed in 2012 by the Wang group, which was based on ZnO nanowire arrays [206].

### Review on Progress and Output Power Optimization in PyENG

Yang et al. [206] developed the first of kind thermal energy harvester based on pyroelectric effect. They used ZnO nanowires in their device, whose pyroelectric and semiconducting properties creates polarization and charge separation along the nanowire due to the time-dependent change in temperature. The pyroelectric current and voltage coefficients are ~1.2–1.5 nC/cm^2^ K and ~2.5–4 × 10^4^ V/mK, respectively. The coefficient of heat conversion into electricity is found to be ~0.05–0.08 Vm^2^/W.

Yang et al. [80] demonstrated a PyENG based on a lead zirconate titanate (PZT) film, which has a pyroelectric coefficient of about −80 nC/cm^2^ K. The developed PyENG, which undergoes a temperature change of 45 K at a rate of ~0.2 K/s, produces an output voltage and current density of ~22 V and ~171 nA/cm^2^. The single output pulse of this PyENG can be used to drive an LCD for more than 60 s. The researchers, through theoretical calculation and analysis, found that the high output of the PyENG depends on the pyroelectric coefficient, the change in temperature, and the thickness of the film.

Ko et al. [207] developed a pyroelectric power generation from relaxor ferroelectric 0.7 Pb (Mg_1/3_Nb_2/3_) O_3_–0.3PbTiO_3_ (PMN-PT) as a single crystal. They can develop the generator near the structural phase–transition temperature. They successfully achieved the open circuit voltage of 1.1 V and 10 nA at room temperature. From their analysis, they were confident about developing a pyro generator for high power applications.

Xue et al. [208] developed a wearable pyroelectric nanogenerator by utilizing the excess heat available naturally from various sources. Mainly, they developed a sensor to analyze the heat fluctuation from human breathing where it recorded at 5 °C ambient temperature. During the process of utilizing the temperature from human breath, it is found that the pyroelectric generator generated an open voltage of 42 V and a short circuit of 2.5 µA. During this process, 50 M ohms is connected and reached up to a maximum of 8.31 microwatts. From the above analysis and working model, they demonstrate the pyroelectric generator can be used as an unencumbered wearable mode, which promises the development of energy harvester for practical applications.

Ma et al. [209] designed an enhanced self-powered UV photoresponse of ferroelectric BaTiO_3_ by the pyroelectric effect. Using the BaTio_3_ developed intrinsic spontaneous polarization with slight temperature fluctuation. They reported a 365 nm UV light photodetector using the light-induced pyroelectric force in Ag/BTO/with the response of 0.5 s at the rising edge. They suggested they can be dramatically enhanced by larger than 1200% with 2.1 K/S temperature variant. So, they proposed a method to improve 365 nm light response by coupling light and heating-induced pyroelectric effect in BTO.

Moalla et al. [210] investigated the monolithically integrated metal insulator metal, which is a 500 nm thick Pb(Zr_0.52_Ti_0.48_)O_3_ structure polycrystalline textured for analyzing the pyroelectric properties. They compared the pyroelectric properties of polycrystalline PZT film and epitaxial PZT film integrated on silicon, both statically (with stabilized temperatures) and dynamically (temperature transient as a pyroelectric generator should work). They calculated the critical difference in the densities of converted pyroelectric energy with almost two magnitudes.

Jiang et al. [211] analyzed the pyroelectric properties of intrinsic GaN nanowires (NW) and nanotubes (NT) based on the size and shape of the semiconductors. The influence of the shape contributes to the significant effect on its potentials by up to dozens of times or even more. For their analysis, they have provided evidence that the pyroelectricity decreases with the inverse of nanocrystal size or with the shape factor.

Raouadi et al. [212] conducted a study and conducted experimental research on harvesting wind energy by pyroelectric nanogenerators (PNG). They demonstrated that using a “PVDF film + vortex generator” PNG could produce an uninterrupted power supply. During the process, they were successful in their results and provided an output current of 0.109 microamperes for 25 m/s wind velocity for a 9 micro meter PVDF film, which was stored in 1 microfarad.

Yang et al. [213] investigated the flexible PbTiO_3_–nanowires for analyzing the dielectric, ferroelectric, and pyroelectric properties. They designed flexible composite thin films with fillers as PbTiO_3_ monocrystalline nanowires, and poly (vinylidene fluoride-tri-fluoroethylene) (P(VDF-TrFE)) as a matrix were prepared. During the process, it is necessary to do hot pressing to eliminate the γ-phase in the solution, which suppresses the switchability of the P (VDF-TrFE) matrix. They found the relative permittivity and dielectric losses of the composites are greatly decreased. Pyroelectric coefficient p of the composites increases with the mass fraction of PbTiO_3_. The voltage FOM F_v_ decreases as the loading of PbTiO_3_ nanowires increases, while detectivity FOM (F_D_) remains relatively high with level for 0–30 wt% loading content. The list of PyENGs developed over the years are listed below in Table 6.

## 7. Conclusions

A comprehensive review of various types of generators and its applications were focused in this paper. The invention of nanogenerators is the most significant milestone in the growing crisis of energy shortage and climate change. Harvesting ambient mechanical energy for electrical and electronics systems are found to be sustainable as they move towards minimization, mobility, and performance. It shows promising potential in the field of transportation, monitoring sensors, biomedical sensors, wind/wave/water drop energy harvesting, rehabilitation devices as smart sensors, etc. The nanogenerators are lightweight, made out of low-cost materials, easily fabricated, are small in size, and above all, give high output intensity, which makes it a promising technology towards sustainability. Nanogenerators can be a successful replacement to batteries in the application of self-powered sensors. Generally, the output voltage of nanogenerators is very high when compared to their output current. From this review, it has been found that among the three types of nanogenerators, the triboelectric nanogenerators are capable of generating higher voltage and power density. This high-voltage low-current issue can be managed by voltage transformers and rectifiers, which can boost the current and reduce the voltage. As the source of power to nanogenerators are intermittent, the output power is not stable, so to provide stable power to electronics, capacitors and highly efficient power management circuits can be used. Through transformers, inductors, and electronic logic control switches, the output power characteristics can be improved. The power density of TENG is also found to be higher when compared with other types of nanogenerators and low-frequency electromagnetic power generators. In the case of blue energy harvesting, efficient system integration with the power management module has to be considered. One of the key concerns of the nanogenerators in blue energy harvesting is their durability. Development of nanogenerators with long durability could be a challenge to researchers. Moreover, nanogenerators can be hybridized with other energy-harvesting devices like electromagnetic generators, solar PV systems, turbines, and energy storage units to increase the overall power conversion efficiency. The problem of higher temperature management in concentrated photovoltaic can be addressed by using pyroelectric nanogenerators. In the past decade, research and real-life applications of nanogenerators have grown exponentially; this shows that in the future, this technology could be widely commercialized.

## Figures and Tables

**Figure 1 nanomaterials-09-00773-f001:**
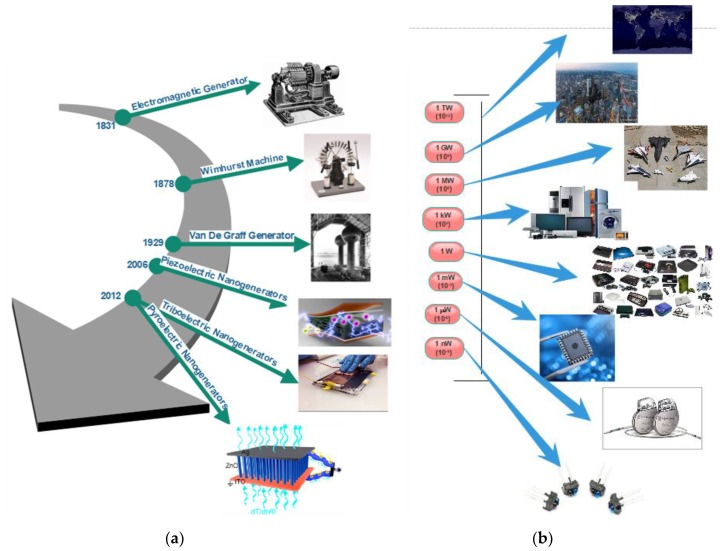
(**a**) The major inventions in the history of mechanical energy-harvesting technology. (**b**) Energy required for various devices at various power scales.

**Figure 2 nanomaterials-09-00773-f002:**
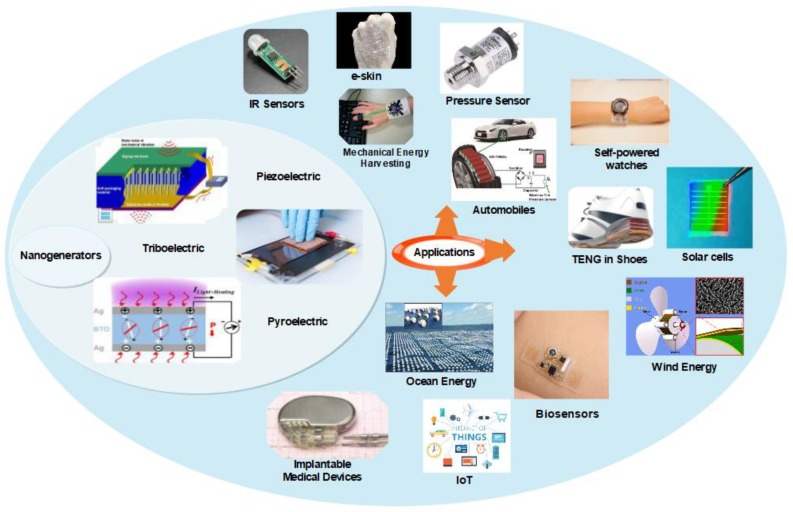
Different types of nanogenerators and their applications in the era of the Internet of Things (IoT).

**Figure 3 nanomaterials-09-00773-f003:**
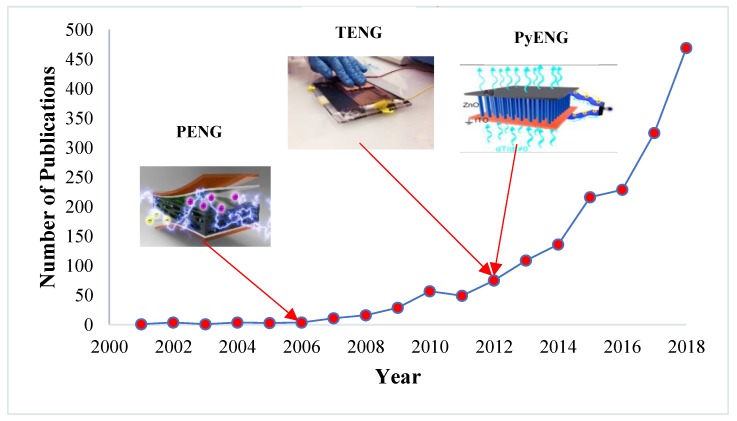
The number publications in the field of nanogenerators over the years.

**Figure 4 nanomaterials-09-00773-f004:**
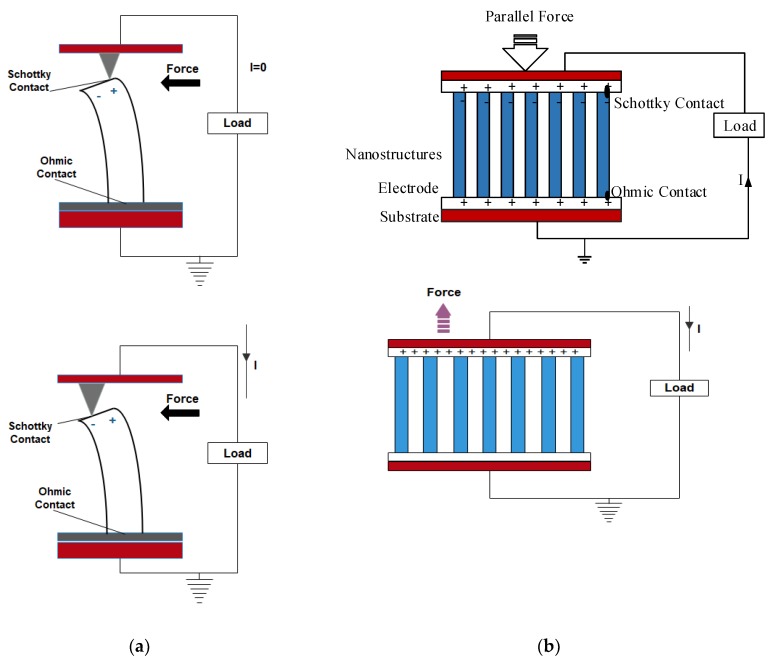
(**a**) Force exerted perpendicular to the growth of the nanowire. *(***b**) Force exerted parallel to the growth of the nanowire.

**Figure 5 nanomaterials-09-00773-f005:**
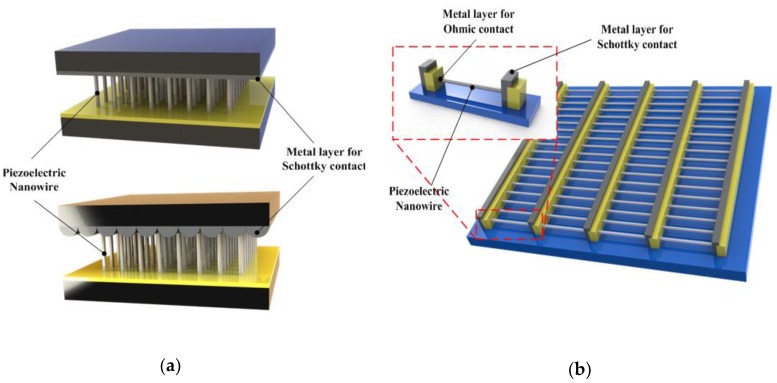
Geometrical configuration of piezoelectric nanogenerator (PENG). *(***a**) Basic structure of vertically integrated nanogenerator. *(***b**) The basic structure of laterally integrated nanogenerator.

**Figure 6 nanomaterials-09-00773-f006:**
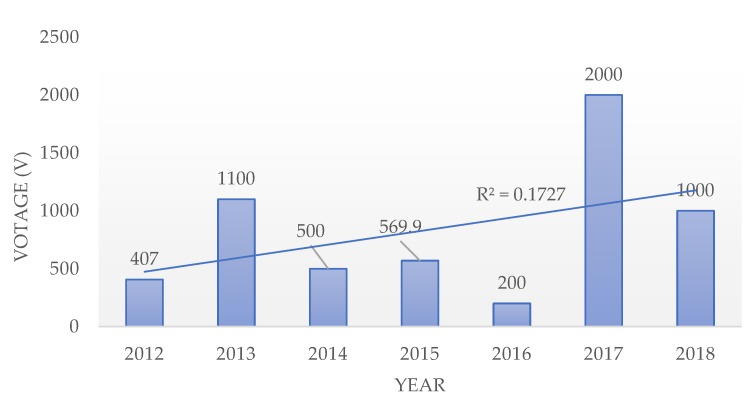
The maximum voltage obtained from TENG over the years.

**Table 1 nanomaterials-09-00773-t001:** Energy potential in human body motions [5].

Body Motions	Mechanical Energy	Available Electrical Energy	Electrical Energy Per Movement
Blood Flow	0.93 W	0.16 W	0.16 J
Exhalation	1 W	0.17 W	1.02 J
Inhalation	0.83 W	0.14 W	0.84 J
Upper Limbs	3 W	0.51 W	2.25 J
Walking	67 W	11.39 W	18.90 J
Fingers Typing	6.9–19 mW	1.2–3.2 mW	226–406 mJ

**Table 2 nanomaterials-09-00773-t002:** Significant improvements in the development of PENG.

Year	Author	Materials	Output Voltage & Short-Circuit Current	Frequency/Strain	Power Output	Output Area Power Density	Output Volume Power Density
2006	Wang et al. [8]	ZnO nanowires	~6–9 mV	~10 MHz	0.5 pW/NW	~1 nW/cm^2^	-
2007	Wang et al. [23]	ZnO nanowires	−0.7 mV, 0.15 nA	41 kHz	~0.1 pW/NW	10 µW/cm^2^	1–4 W/cm^3^
2008	Qin et al. [24]	ZnO nanowires	1–3 mV, 4 nA	<10 Hz	-	20–80 mW/cm^2^	-
2009	Yang et al. [26]	ZnO nanowires/Kapton film	~50 mV, 400–750 pA.	22 cycles per minute	-	-	
2010	Zhu et al. [27]	ZnO nanowires/PDMS film/Au film	2.03 V, 107 nA, 200 pA (single nanowire)	0.33 Hz, 0.1% strain, and strain rate of 5% s^−1^.	-	22 µW/cm^2^ (single layer)0.44 mW/cm^2^ (20 NW layers)	~11 mW/cm^3^ (single layer)~1.1 W/cm^3^ (20 NW layers)
2010	Xu et al. [29]	ZnO nanowires	1.26 V, 28.8 nA	0.19% strain	-	-	2.7 mW/cm^3^
2010	Huang et al. [30]	InN nanowires	1 V	-	-	-	-
2010	Chen et al. [31]	PZT nanofibres/platinum wires/PDMS	1.63 V	39.8 Hz	0.03 µW		
2011	Cha et al. [32]	PVDF nanowires	2.6 V, 0.6 µA	-	-	-	0.17 mW/cm^3^
2012	Zhu et al. [18]	ZnO nanowires/PMMA layer/ITO layer/Al	58 V, 134 µA	-	-	-	0.78 W/cm^3^
2012	Hu et al. [35]	ZnO nanowires/PMMA/Cr, Au electrodes	20 V, 6 µA	0.12% strain at strain rate of 3.56% s^−1^.	-	-	0.2 W/cm^3^
2016	Ghosh and Mandal [44]	Transparent fish scale	4 V, 1.5 µA	0.17 MPa	-	1.14 µW/cm^2^	-
2016	Lu et al. [43]	ZnO/PMMA/FTO, gold electrodes	2 V	-	-	-	-
2017	Cho et al. [42]	Li-doped CuO_2_/ZnO	-	-	~52.5 µW	-	-
2017	Haibo et al. [45]	NKN nanorods	35 V, 5 µA	2.13% strain at strain rate of 3.7% s^−1^.	16.5 µW	-	-
2017	Chen et al. [46]	P(VDF-TrFE)/BaTiO_3_	13.2 V	-	-	-	-
2017	Ku et al. [56]	InN	825 µV	5 Hz	-	2.9 nW/cm^2^	-
2017	Kang et al. [53]	GaN/PET	4.2 V, 150 nA	Shear stress ~182 mN	-	-	-
2018	Dudem et al. [47]	BaTiO_3_/PVDF, Ag/BTO	14 V, 0.96 µA	3 N, 5 Hz	-	~98.6 µW/cm^2^	-
2018	Jenkins et al. [49]	FF peptide nanowire	−0.6 V, 7 nA	10 nN force applied	0.1 nW	-	-
2018	Johar et al. [54]	p-n NiO/GaN/PDMS/ITO/PET	30 V, 1.43 µA	20 Hz	-	-	-
2018	Johar et al. [55]	GaN/Ni/PDMS	15 V, 85 nA	-	-	-	-
2018	Lee et al. [57]	PZT-NH_2_ nanoparticels	65 V, 1.6 µA	-	26 µW	-	-
2019	Filippin et al. [51]	ZnO nanowires	170 mV	-	-	-	-
2019	Maria et al. [58]	Bi_4_Ti_3_O_12_, BiTO NPs/PDMS	12.5 V, 100 nA	-	-	562 µW/cm^2^	-

**Table 3 nanomaterials-09-00773-t003:** Comparison of the working principle of various modes of operation of TENG.

Modes of Operation	Mechanism	Application
1. Vertical-contact separation mode 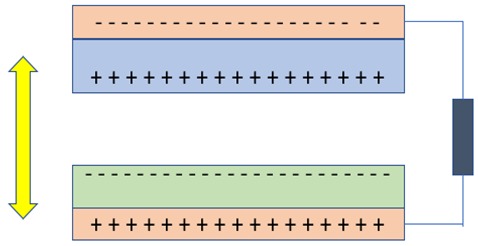	The first invented operation mode in TENGs. The external force is applied vertically on the triboelectric materials. The charges are formed on the surface of the materials, and as they are separated by a distance, an electric potential difference is established between the two electrodes attached to the triboelectric materials. The reciprocal motion of the materials can produce alternating current.	To harvest mechanical energy from human motions, the vibration of a machine, wind, flowing water, etc.
2. In-plane sliding mode 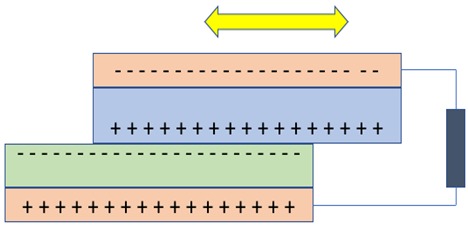	The two triboelectric surfaces slide in the lateral direction. A lateral polarization is introduced between the materials that push the electrons from the top electron to the bottom electrode to balance the induced potential. The periodic sliding and closing can generate alternating power. The sliding can be planar, cylindrical/disc rotation.	Used to generate power from rotational motion, waves, pressing/touching, etc.
3. Single-Electrode mode 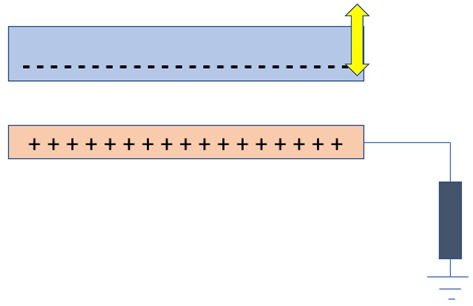	Used to harvest energy from arbitrarily freely moving objects, unlike in the other two operation mode. It has a triboelectric surface and only one electrode and is grounded, as shown in the figure. The change of the distance between the two surfaces causes charge transfer between the electrode and the ground, thus driving electricity through an external load.	Used to harvest energy from mobile objects. Can harvest energy from turning book pages, raindrops, rotating tire, footsteps, etc.
4. Free-standing mode 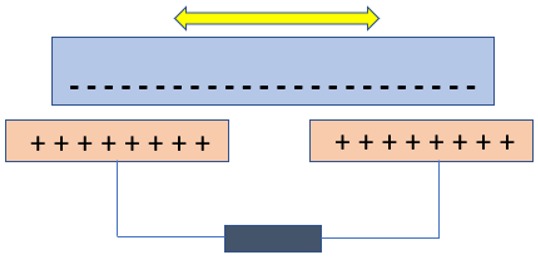	Used to harvest energy from moving objects. A pair of identical electrodes are placed below the triboelectric layer with a gap distance. An asymmetric charge distribution is generated in the media as the triboelectric layer is brought in contact and separated from the electrodes. Due to this, the electrons will flow between the electrodes to equate the potential distribution. Since there is no contact between the triboelectric layer and the electrodes, there are no chances of wear and tear. High power conversion efficiency compared to other modes of operation.	Harvests energy from automobiles, human walking, air flow, computer mouse operation, etc.

**Table 4 nanomaterials-09-00773-t004:** Triboelectric series for some commonly available materials [95].

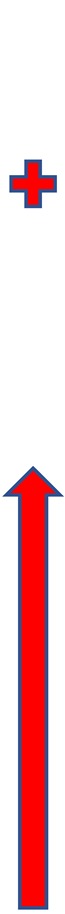	Aniline-formol resin	Polyvinyl alcohol	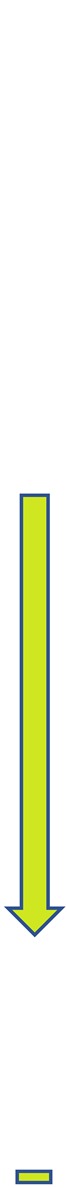
Polyformaldehyde 1.3–1.4	Polyester (Dacron) (PET)
Ethylcellulose	Polyisobutylene
Polyamide 11	Polyuretane flexible sponge
Polyamide 6-6	Polyethene terephthalate
Melanie formol	Polyvinyl butyral
Wool, knitted	Formo-phenolic, hardened
Silk, woven	polychlorobutadiene
Polyethene glycol succinate	Butadiene-acrylonitrile copolymer Nature
Cellulose	Nature rubber
Cellulose acetate	Polyacrylonitrile
Polyethene glycol adipate	Acrylonitrile-vinyl chloride
Polydiallyl phthalate	Polybisphenol carbonate
Cellulose (regenerated) sponge	Polychloroether
Cotton, Woven	Polyvinylidene chloride (Saran)
Polyurethane elastomer	Poly(2,6-dimethyl polyphenyleneoxide)
Styrene-acrylonitrile copolymer	Polystyrene
Styrene-butadiene copolymer	Polyethylene
Wood	Polypropylene
Hard rubber	Polydiphenyl propane carbonate
Acetate, Rayon	Polyimide (Kapton)
Polymethyl methacrylate (Lucite)	Polyethylene terephtalate
Polyvinyl alcohol	Polyvinyl Chloride (PVC)
(continued)	Polytrifluorochloroethylene
Polyamide 11	Polyisobutylene
Polyamide 6-6	Polyuretane flexible sponge
Melanime formol	Polyethylene Terephthalate
Wool, knitted	Polyvinyl butyral
Silk, woven	Polychlorobutadiene
Aluminum	Natural rubber
paper	Polyacrilonitrile
Cotton, Woven	Acrylonitrile-vinyl chloride
Steel	Polybisphenol carbonate
Wood	Polychloroether
Hard rubber	Polyvinylidine chloride (Saran)
Nickel, copper	Polystyrene
Sulfur	Polyethylene
Brass, silver	Polypropylene
Acetate, Rayon	Polyimide (Kapton)
Polymethyl methacrylate (Lucite)	Polyvinyl Chloride (PVC)Polyvinyl
Polyvinyl alcohol	Polydimethylsiloxane (PDMS)
(continued)	Polytetrafluoroethylene (Teflon)

**Table 5 nanomaterials-09-00773-t005:** Summary of experimental studies on different triboelectric nanogenerators (TENGs).

Year	Authors	Tribo-Layer Used in TENG	The Electrode Used in TENG	Open-Circuit Voltage (V_OC_) (V)	Short-Circuit Current (I_SC_) (µA)	Current Density	Surface Power Density	Power Density and Power	Load Resistance (Ω)	Applications
2012	Fan et al. [88]	PET/Kapton	Gold(Au)–Palladium(Pd) alloy film	3.3	0.6	-	-	~10.4 mW/m^3^	-	Self-powered systems
2012	Zhu et al. [98]	PMMA/Kapton nanowires	Aluminum	110	6	-	-	31.2 mW/m^3^	-	Small portable electronics
2012	Wang et al. [186]	PDMS/Kapton/SiO_2_	Aluminum	230	94	-	-	128 mW/m^3^	50 MΩ	Portable electronics
2012	Zhong et al. [187]	PTFE/PET	Cu/Ag	407	~33	-	-	~4.125 mW	-	Mobile electronics
2013	Zhu et al. [188]	Acrylic/PTFE	Aluminum/Copper	615	0.44	0.18 A/m^2^	^-^	^-^	-	Mechanical energy harvesting
2013	Hu et al. [101]	Acrylic sheet/Kapton film	Al/Cu	110	15	-	2.76 W/m^2^	-	6 MΩ	Wave energy harvesting
2013	Chen et al. [102]	PTFE/Acrylic	Al/Cu	287.4	76.8	-	726.1 mW/m^2^	-	-	Self-powered vibration sensors
2013	Yang et al. [112]	PTFE/PET	Al/Cu	428	1.395	-	30.75 W/m^2^	-	2 MΩ	Harvest walking energy
2013	Cheng et al. [103]	PMMA/PDMS/SiO_2_ nanoparticles	Au film	285	0.53	1325 A/m^2^	3.6 × 10^5^ W/m^2^	142 W	500 MΩ	High-pulsed power source
		PMMA/PDMS/SiO_2_ nanoparticles	Au film	115	5.2	104 A/m^2^	1.4 × 10^4^ W/m^2^	-	22 MΩ	Self-powered systems
2013	Hou et al. [142]	PET/PDMS	Indium tin oxide (ITO)/Cu	220	40	0.8 µA/m^2^	-	-	10^3^–10^8^ Ω	Harvesting walking energy for sensors
2013	Yang et al. [189]	PTFE	Al	1100	-	6 mA/m^2^	350 mW/m^2^	-	100 MΩ	Self-powered sensors
2014	Zhu et al. [115]	PTFE	Metal gratings	0–500	-	-	500 W/m^2^	15 × 10^6^ W/m^3^, 3 W(average power)	-	Self-powered electronics
2014	Lin et al. [105]	PTFE	PMMA/Cu	9.3	17	-	200 W/m^2^	-	5 MΩ	Water drop energy
2014	Liang et al. [190]	PTFE	ITO	10	-	-	11.56 mW/m^2^	-	0.5 MΩ	Energy from water drops
2014	Zheng et al. [180]	PTFE	ITO/PET	30	-	4.2 mA/m^2^	-	-	-	Solar PV
2014	Yang et al. [135]	Acrylic/polyolefin	Al	−1070	^-^	10 mA/m^2^	288 mW/m^2^	-	100 MΩ	e-skin
2015	Sun et al. [191]	Ethyl cellulose/polylactic acid/Kapton	Copper wire/silver paste	310.5	16.2	-	-	-	0–200 KΩ	Harvesting energy from biological activities
2015	Huang et al. [192]	Polyvinylidene fluoride/poly(3-hydroxybutyrate-co-3-hydroxyvalerate)/PET substrate	Aluminum	340	78	-	2.3 W/m^2^	-	0.1–40 MΩ	Self-powered sensors
2015	Chen et al. [159]	Acrylic/PET/PTFE	Al/Cu	569.9	930	-	2.6 W/m^2^	-	1–10 MΩ	Blue energy harvesting
2015	Zhang et al. [193]	PET	ITO	98	16.3	-	2.76 W/m^2^	-	-	Wind energy harvesting
2015	Wu et al. [194]	PET/PDMS/rice husk/PTFE	Al/Cu	270	14	5.7 mA/m^2^	0.84 W/m^2^	-	200 MΩ	Industrial applications
2015	Wang et al. [156]	Nylon 6/6 and Kapton	Al	~900	1	-	-	10 mW	10 GΩ	Water wave energy harvesting
2016	Park et al. [195]	PTFE	Al/Cu nanostructures	200	~10	-	-	-	-	Self-powered sensors
2016	Chen et al. [196]	PTFE	Al	150	78	-	-	8.58 mW	6 MΩ	Force sensing
2016	Yong et al. [197]	Expanded polystyrene (EPS)/Polyvinyl Chloride (PVC)	Ag	11.2	1.86	-	-	-	-	Wind energy harvesting
2017	Chen et al. [107]	Printed composite resin	ionic hydrogel	62	-	-	-	10.98 W/m^3^	-	Wearable devices/AI/IoT
2017	Mallineni et al. [97]	Kapton/Teflon	gPLA	2000	-	-	-	70 mW	-	Self-powered sensors
2017	Li et al. [168]	FEP nanowires/Au	Ag	200	10	-	-	-	-	Blue energy/wireless infrared system
2018	Feng et al. [96]	PDMS/polyamide 6/leaf powder/poly-l-lysine to modify the surface	Conductive double-sided carbon	1000 (max)	150 (max)	-	-	-	-	Wind energy harvesting
2018	Qian et al. [140]	Polymide/PDMS/PTFE	aluminim	316	-	-	-	22.3 mW	3 MΩ	Pressure monitoring
2018	Liu et al. [182]	PDMS	PEDOT:PSS	~2.14	~0.033	-	17.4 W/m^2^	-	-	Solar PV
2018	Dong et al. [120]	MXene	PET-ITO	~500 to ~650	-	-	-	0.5–065 mW	-	wearable/flexible electronics
2018	Lei et al. [162]	PTFE	Cu	707.01	75.35	-	-	9.559 W/m^3^	-	Blue energy harvesting
2018	Pang et al. [161]	Calcium alginate	Al	33	0.15			9.5 µW		Blue energy harvesting
2018	Sun et al. [134]	PDMS	Ag nanowires/PEDOT	~100	-	-	3200 W/m^2^	-	-	Energy skin/soft devices
2019	Luo et al. [124]	FEP/PDMS/Kapton/PET	Al	-	37	-	-	1.83 mW	2 MΩ	Flexible electronics
2018	Wang et al. [198]	Silicone/Kapton	Cu	290 150	2.8 15	-	-	165 µW 850 µW	20 MΩ 100 Ω	Desalination/self-powered marine rescue system
2019	Xia et al. [199]	Solid milk film/PTFE film	Conductive ink	392	93	-	5.837 W/m^2^	-	-	Flexible, wearable electronics
2019	Hao et al. [175]	PMMA/silicon (TENG/EMG hybrid)	Ag/Al	80 V/13 V	1.2 µA/2000 µA	-	-	TENG: 0.08 mW EMG: 179 mW	100 MΩ 1 kΩ	

**Table 6 nanomaterials-09-00773-t006:** Significant improvements in the development of pyroelectric nanogenerator (PyENG).

Year	Authors	Pyroelectric Material and Electrodes	Pyroelectric Current Coefficient	Pyroelectric Voltage Coefficient	Output Voltage/Current	Current Density	Power Density/Power
2012	Yang et al. [206]	ZnO nanowires, Ag/ITO	~1.2–1.5 nC/cm^2^ K	~2.5–4 × 10^4^ V/mK	-	-	~0.05–0.08 Vm^2^/W
2012	Yang et al. [80]	PZT film, Cu/Ni layer	−80 nC/cm^2^ K	-	~22 V	171 nA/cm^2^	
2014	Ko et al. [207]	PMN-PT	~104–235 nC/cm2 K	-	1.1 V, 10 nA	-	-
2017	Xue et al. [208]	PVDF the film, Al	27 µC/m^2^ K	-	42 V, 2.5 µA	-	8.31 µW
2017	Ma et al. [209]	BaTio_3_, Ag	2.1 nC/cm^2^ K	-	2.2 nA	-	60.3 nW
2017	Moalla et al. [210]	Pb(Zr_0.52_ Ti_0.48_)O_3_	−470 µC/m^2^ K (static) 30 µC/m^2^ K (dynamic)	-	-	-	-
2017	Jiang et al. [211]	GaN	-	7 × 10^5^ V/mK	-	-	-
2017	Raouadi et al. [212]	PVDF film/vortex generator	27.15 µC/m^2^ K	-	-	0.109 µA/cm^2^	2.82 µW/cm^2^
2018	Yang et al. [213]	PbTiO_3_ nanowires/P(VDF-TrFE)	52.7 µC/m^2^ K/72.8 µC/m^2^ K	-	-	-	-

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
