# Peer review of "Nanogenerators as a Sustainable Power Source: State of Art, Applications, and Challenges"

_nanomaterials, 2019, doi:10.3390/nano9050773_

Reviewer 1 Report

Authors have given a very detailed survey of latest work in the cited arena. According to me, it will be of importance and valuable to the readers. I recommend its publication in the present format.

Author Response

Dear Reviewer,

Thank you for your positive feedback on our manuscript and recommending its publication in its present format.

Sincerely,

Sridhar S I1, Aravind C V2*, Kameswara Satya Prakash O3, Faizal Fazuan4, and Saidur R5

1,2*,3,4 School of Engineering, Faculty of Innovation and Technology, Taylor’s University Lakeside Campus, No. 1, Jalan Taylor’s 47500 Subang Jaya, Selangor, Malaysia, 5Research Centre for Nano-Materials and Energy Technology, Sunway University, Malaysia.

*   Correspondence to aravindcv@ieee.org

Reviewer 2 Report

The manuscript “Nanogenerators as a Sustainable Power Source: State of Art, Applications and Challenges” by Sridhar S I et al is a review dealing with the piezoelectric, triboelectric, thermoelectric and pyroelectric nanogenerators view as a suitable energy sources.

On the subject of nanogenerators, many reviews are already published, too often with similar data, results...

A review have to be constructed for the reader by discussing specific point supported by references.

In this paper, the authors groups together several types of nanogenerators, but without discussing in details each parts and being exhaustive. The following remarks support my argumentation:

In the parts dealing with the description of the effect in play in each type of nanogenerators, the authors are not sufficiently precise for a review. For example, in the case of piezoelectric effect for PENG, the creation and distribution of the piezoelectric field inside 1D-nanostructures, which is strongly link to the polarity of the materials and the type of mechanical deformation, is not explained. The figure 4a) is not conformed since they need to schematize a laterally bended nanowires. In addition, the authors describe only the case where the piezoelectric charges are collected through a Schottky contact, while in the referred papers, the capacitive configuration is also considered.

The authors are comparing results that are not comparable. In fact, the load resistances used to measure the output signal are generally not the same, while this parameter affect strongly the measured signal. In this view, the Figure 4 (the second, page 20) is not constructive. In addition, in the different tables, the unities are not the same (mW/cm², mW/m² for exemple !). Finally, the conditions of deformations are not always indicated, which do not allow a good understanding of the progress in this field. It is true that in the nanogenerator community, there is no consensus about the measurements and the unity, which induces complexities to discuss and compare the results. However, the authors discussed the progress of the output power generated. At minimum, this problem of comparison must be discussed.

To address the progress, the authors list several publications. They do not construct the review by addressing the main point of materials, design optimization. In addition, the electrode used to collect the electrical charges is of great importance in the device performances. This point is not discussed in this review.

A review is not a simple list of publications with some details. The authors have made this choice without schemas to support the description. This does not make simple reading. In addition, the lists of publication are not exhaustive. This is particularly true in the case of PENG. By reading this review, we can have the idea that only ZnO and few 1D nanostructures present potentiality to develop PENG. No reference to PZT, GaN, InN, CdS, CdSe … 1D-nanostructures while they present high piezoelectric response.

In conclusion, the proposed review cannot be accepted for publication.

Author Response

Dear Reviewer,

Thank you for the thoughtful and constructive feedback you provided regarding our manuscript. We have amended this manuscript based on your comments and suggestions.The rebuttal is attached with this letter. We are certain that you will find this revised version of our manuscript clears up the issue you indicated in your response.

With these changes to our manuscript, we hereby resubmit our manuscript for a secondary evaluation. Thank you once again for your consideration of our paper.

Sincerely,

Sridhar S I1, Aravind C V2*, Kameswara Satya Prakash O3, Faizal Fazuan4, and Saidur R5

1,2*,3,4 School of Engineering, Faculty of Innovation and Technology, Taylor’s University Lakeside Campus, No. 1, Jalan Taylor’s 47500 Subang Jaya, Selangor, Malaysia, 5Research Centre for Nano-Materials and Energy Technology, Sunway University, Malaysia.

*   Correspondence to aravindcv@ieee.org

Reviewer 3 Report

   This article presents an informative and well-summarized review regarding the development of nanogenerators in a recent year. The submitted manuscript is well organized with full of references. Therefore, the original manuscript is proper to be published; however, it may need to fix some issues before publication. The followings are a list for that.

Figure 2 seems to me not clear, so that I could not catch the applications. It should be re-draw with some words for each technology representing.

At the first paragraph of Chapter 3, the explanation of piezoelectric generation does not match with common definition of piezoelectricity. Refer to the definition of piezoelectricity, explanation of piezoelectric generation should be addressed correctly. Furthermore, the insertion of explanation regarding the role of Schottky barrier would be very helpful to the reader’s understanding.

There are many typos in the manuscript, especially in the numerical values and units. It needs to be revised thoroughly.

In Table 2, the units of output power density are needed to be unified at least the dimension of volume or surface like cm3 or cm2. The current various units of power density do not give any of informative values to the reader.

In Table 3, some explanation in Mechanism section did not matched with its mode. It should be revised thoroughly.

A explanation of fitting line in Figure 4 is missing.

Author Response

Dear Reviewer,

Thank you for the thoughtful and constructive feedback you provided regarding our manuscript. We have amended this manuscript based on your comments and suggestions. The rebuttal is attached with this letter. We are certain that you will find this revised version of our manuscript clears up the issue you indicated in your response.

With these changes to our manuscript, we hereby resubmit our manuscript for a secondary evaluation. Thank you once again for your consideration of our paper.

Sincerely,

Sridhar S I1, Aravind C V2*, Kameswara Satya Prakash O3, Faizal Fazuan4, and Saidur R5

1,2*,3,4 School of Engineering, Faculty of Innovation and Technology, Taylor’s University Lakeside Campus, No. 1, Jalan Taylor’s 47500 Subang Jaya, Selangor, Malaysia, 5Research Centre for Nano-Materials and Energy Technology, Sunway University, Malaysia.

*   Correspondence to aravindcv@ieee.org

Round  2

Reviewer 2 Report

After the first revision of the  manuscript, several points need to be improved again, especially the point discussing the piezoelectric generation mechanisms, where information are missing and mistakes can be found. A major revision, concerning the part 3, is needed before that paper should be considered for publication.

The argumentation gives in page 5 There are two cases of PENG[11,16], one is where the individual nanowire/nanorod[17] is subjected to the strain exerted perpendicular to the growing direction of the nanowire/nanorod which leads to the generation of the electric field. Figure 4a shows the working of PENG when the force is applied perpendicular to its axis. When force is applied one portion of the tip of the nanowire undergoes expansion, and the other undergoes compression. The compressed portion exhibits a positive strain and positive potential while the expanded portion exhibits negative strain and negative potential; hence the electric field is distributed at the ends of the nanowire. This electric field gets neutralized by the ohmic contact formed between the bottom electrode and the nanowire. The Schottky contact between the top electrode and the nanowire generates electric current because of its rectifying characteristics [8].” is correct in the specific case of n-doped ZnO NWs (Zn2+ polarity). It doen’t work for p-doped ZnO NWs, or nanowires presenting anion polarity such as III-N NWs. The authors must specify the case in which the described mechanisms can be applied. In addition, the figure 4(a) do not represent correctly the described mechanism. The NWs has to be bended and the direction of the force applied by the AFM tip has to be represented.

The argumentation gives in page 5 concerning the second case of applied force “The other case where the external strain is exerted parallel to the growing direction of the nanowire/nanorod(Figure 4b). When the force is applied to the tip of the laterally grown nanowire which stacked between the Schottky contact and ohmic contact a uniaxial compressive is generated in the nanowire. The tip of the nanowire will have negative potential and increases the Fermi level due to the piezoelectric effect. As the electrons flow from the tip of the nanowire to the bottom through the external circuit positive potential will be generated at the tip. The Schottky contact blocks the flow of electrons through the nanowires instead passes the electrons through the external circuit. When the applied force is removed the piezoelectric effect diminishes immediately a positive potential at the tip gets neutralized because of the migration of electrons from the bottom electrode to the top which produces voltage peak in the opposite direction hence alternating current (AC) is generated.” Is incorrect. First, when the NWs are submitted to a compressive strain along their axis, the top of NWs are subjected to compression and thus a positive piezoelectric potential is created (by regarding the description for the first case, when a force is applied perpendicularly to the NWs), and not a negative one as described. The authors have once again to details the characteristics of the NWs considered here and specifically the polarity of the NWs which is well known to determine the piezoelectric potential distribution inside the NWs as a function of their deformation (compression or tensile). Second, and the main mistake, is that when a Schottky diode is used to harvest the piezoelectric output signal generated by the NWs, there is never an AC signal. If the people characterizing nanogenerator device with Schottky contact are measuring an AC signal, it means that the Schottky diode is not effective. It plays the role of a series resistance. Only for capacitive configuration and the case where the Schottky diode is a series resistance, we can observe AC signal. When the Schottky diode is effective (the case that must be described by the authors in this review), the signal is collected only when the diode is positively bias. The problem is that in the literature, several characterizations of nanogenerators with Schottky contact are measuring AC signal, without discussing the real behavior of the Schottky diode and its role. A review cannot describe incorrect behavior. The authors must correct their description and not base it only on literature observations without any critical viewpoint.

The description in part 3 only addresses the case where the NWs are vertically integrated. The case when the NWs are horizontally integrated is not discussed. This point is missing in the kind of review, where the authors describe the piezoelectric mechanisms and take as example vertically and horizontally integration.

In the part 3.1 Progress and Output Power Optimization in PENGs, the authors are listing several results without any organization, in the sense where they address both results obtained on single NWs or 1D-nanostructures and on devices integrating nanostructure arrays. This methodology gives some confusions and is not constructive for a review. A distinction between single nanostructure characterizations and device characterization must be implemented.

In addition, the following argumentation is not acceptable for a review: “This is avoided in the nanotechnology community to nullify the complexities in discussing and comparing the results.” It is not so complex! In reality, it is not a will of a part of the community to compare the results. For each people characterizing piezoelectric properties of a device, it is simple to calculate the power density, which can be easily compare with the literature. But the people don’t want to give this value, because, it is more impressive to give high output voltage for large surface, than that a small output voltage for a short surface, while the power density will be better in the second case. To illustrate this argument, I can give the following example: The authors have added references for GaN NWs. They review the work of the Prof. Ryu, while their systems present power density lower than the state of the art for this class of materials, state of the art settles by a French group (power density reaching up to several tens of mW/cm3).

So to discuss the fact that the community has not yet established conventions to compare their results, the authors must give another argumentation than the one cited before.

Author Response

Dear Reviewer,

Thank you for the thoughtful and constructive feedback you provided regarding our manuscript. We agree with you and We have made the changes accordingly. The responses are mentioned in the attachment. We hope that you will find this most recent version of our manuscript clears up the main issues you indicated in your response. With these changes we hereby resubmit our manuscript for evaluation.

Thank you once again for your consideration of our paper.

Sincerely,

Sridhar S I1, Aravind C V2*, Kameswara Satya Prakash O3, Faizal Fazuan4, and Saidur R5

1,2*,3,4 School of Engineering, Faculty of Innovation and Technology, Taylor’s University Lakeside Campus, No. 1, Jalan Taylor’s 47500 Subang Jaya, Selangor, Malaysia, 5Research Centre for Nano-Materials and Energy Technology, Sunway University, Malaysia.

*   Correspondence to aravindcv@ieee.org
